# COVID-19 in Patients with Inflammatory Bowel Disease: The Israeli Experience

**DOI:** 10.3390/vaccines10030376

**Published:** 2022-02-28

**Authors:** Lev Lichtenstein, Benjamin Koslowsky, Ami Ben Ya’acov, Irit Avni-Biron, Baruch Ovadia, Ofer Ben-Bassat, Timna Naftali, Uri Kopylov, Yael Haberman, Hagar Banai Eran, Rami Eliakim, Adi Lahat-Zok, Ayal Hirsch, Eran Zittan, Nitsan Maharshak, Matti Waterman, Eran Israeli, Idan Goren, Jacob E. Ollech, Henit Yanai, Bella Ungar, Benjamin Avidan, Dana Ben Hur, Bernardo Melamud, Ori Segol, Zippora Shalem, Iris Dotan, Selwyn H. Odes, Shomron Ben-Horin, Yf’at Snir, Yael Milgrom, Efrat Broide, Eran Goldin, Shmuel Delgado, Yulia Ron, Nathaniel Aviv Cohen, Eran Maoz, Maya Zborovsky, Safwat Odeh, Naim Abu Freha, Eyal Shachar, Yehuda Chowers, Tal Engel, Hila Reiss-Mintz, Arie Segal, Adar Zinger, Ariella Bar-Gil Shitrit

**Affiliations:** 1Clalit Health Services, Tel Aviv, Israel; levli@clalit.org.il (L.L.); eranma@clalit.org.il (E.M.); mayazb@clalit.org.il (M.Z.); 2Digestive Diseases Institute, Shaare Zedek Medical Center, Faculty of Medicine, Hebrew University of Jerusalem, Jerusalem, Israel; binyaminkos@szmc.org.il (B.K.); amib@szmc.org.il (A.B.Y.); erang@szmc.org.il (E.G.); 3Division of Gastroenterology, Rabin Medical Center, Petah Tikva, Israel; iritab@clalit.org.il (I.A.-B.); agarb@clalit.org.il (H.B.E.); goreni@ccf.org (I.G.); jacobel@clalit.org.il (J.E.O.); henitya@clalit.org.il (H.Y.); irisdo@clalit.org.il (I.D.); yifats3@clalit.org.il (Y.S.); 4Sackler Faculty of Medicine, Tel Aviv University, Tel Aviv, Israel; timna.naftali@clalit.org.il (T.N.); uri.kopylov@sheba.health.gov.il (U.K.); yael.haberman@sheba.health.gov.il (Y.H.); abraham.eliakim@sheba.health.gov.il (R.E.); adi.lahat@sheba.health.gov.il (A.L.-Z.); ayalh@tlvmc.gov.il (A.H.); nitsanm@tlvmc.gov.il (N.M.); erani@wmc.gov.il (E.I.); bela.gayshis@sheba.health.gov.il (B.U.); avidanb@sheba.health.gov.il (B.A.); bernardom@wmc.gov.il (B.M.); shalemt@clalit.org.il (Z.S.); odes@bgu.ac.il (S.H.O.); shomron.benhorin@sheba.health.gov.il (S.B.-H.); efratbro@shamir.gov.il (E.B.); yuliar@tlvmc.gov.il (Y.R.); nathaniel@tlvmc.gov.il (N.A.C.); eyal.shahar@sheba.health.gov.il (E.S.); tal.engel@sheba.health.gov.il (T.E.); hillar@mhmc.co.il (H.R.-M.); 5Department of Gastroenterology and Hepatology, Hillel Yaffe Medical Center, Hadera, Israel; barucho@hy.health.gov; 6Barzilai Medical Center, Ashkelon, Israel; oferbe@bmc.gov.il; 7Department of Gastroenterology and Liver Diseases, Meir Medical Center, Kfar Saba, Israel; 8Department of Gastroenterology, Chaim Sheba Medical Center, Tel Hashomer, Ramat Gan, Israel; 9Sourasky Medical Center, Tel Aviv, Israel; 10Inflammatory Bowel Disease Unit, Ha’emek Medical Center, Faculty of Medicine, Israel Institute of Technology, Afula, Israel; eranzittan@clalit.org.il; 11Faculty of Medicine, Israel Institute of Technology, Haifa, Israel; m_waterman@rambam.health.gov.il (M.W.); d_ben-hur@rambam.health.gov.il (D.B.H.); y_chowers@rambam.health.gov.il (Y.C.); 12Rambam Medical Center, Faculty of Medicine, Tel Aviv University, Tel Aviv, Israel; 13Department of Gastroenterology and Liver Diseases, Wolfson Medical Center, Holon, Israel; 14Unit of Gastroenterology, Lady Davis Carmel Medical Center, Haifa, Israel; ori_segol@clalit.org.il; 15Gastroenterology and Liver Diseases Institute, Shamir Medical Center, Be’er Ya’akov, Israel; 16Mayanei HaYeshua Medical Center, Bnei Brak, Israel; 17Hadassah Medical Center, Jerusalem, Israel; yaelmil@hadassah.org.il (Y.M.); adar@hadassah.org.il (A.Z.); 18Assuta Medical Center, Ben-Gurion University, Negev, Be’er Sheva, Israel; delgado@bgu.ac.il; 19Bnai Zion Medical Center, Haifa, Israel; safwat.odeh@b-zion.org.il; 20Soroka Medical Center, Be’er Sheva, Israel; naimf@clalit.org.il (N.A.F.); arikse@clalit.org.il (A.S.)

**Keywords:** COVID-19, Crohn’s disease, ulcerative colitis, inflammatory bowel disease, biological drugs, immune suppression

## Abstract

Background: Crohn’s disease (CD) and ulcerative colitis (UC) are chronic, immune-mediated inflammatory bowel diseases (IBD) affecting millions of people worldwide. IBD therapies, designed for continuous immune suppression, often render patients more susceptible to infections. The effect of the immune suppression on the risk of coronavirus disease-19 (COVID-19) is not fully determined yet. Objective: To describe COVID-19 characteristics and outcomes and to evaluate the association between IBD phenotypes, infection outcomes and immunomodulatory therapies. Methods: In this multi-center study, we prospectively followed IBD patients with proven COVID-19. De-identified data from medical charts were collected including age, gender, IBD type, IBD clinical activity, IBD treatments, comorbidities, symptoms and outcomes of COVID-19. A multivariable regression model was used to examine the effect of immunosuppressant drugs on the risk of infection by COVID-19 and the outcomes. Results: Of 144 IBD patients, 104 (72%) were CD and 40 (28%) were UC. Mean age was 32.2 ± 12.6 years. No mortalities were reported. In total, 94 patients (65.3%) received biologic therapy. Of them, 51 (54%) at escalated doses, 10 (11%) in combination with immunomodulators and 9 (10%) with concomitant corticosteroids. Disease location, behavior and activity did not correlate with the severity of COVID-19. Biologics as monotherapy or with immunomodulators or corticosteroids were not associated with more severe infection. On the contrary, patients receiving biologics had significantly milder infection course (*p* = 0.001) and were less likely to be hospitalized (*p* = 0.001). Treatment was postponed in 34.7% of patients until recovery from COVID-19, without consequent exacerbation. Conclusion: We did not witness aggravated COVID-19 outcomes in patients with IBD. Patients treated with biologics had a favorable outcome.

## 1. Introduction

Inflammatory bowel disease (IBD) is a complex, multifactorial chronic inflammatory disease of the gastrointestinal tract encompassing two main clinical entities: Crohn’s disease (CD) and ulcerative colitis (UC). Given the immunological signature of IBD, the current treatment strategies frequently involve immunosuppressive drugs that are aimed at controlling the excessive intestinal immune response. The extensive use of these therapies among patients with IBD increases their risk of opportunistic and severe infections [1,2]. Patients with IBD are among millions of people that have been affected by the novel severe acute respiratory syndrome coronavirus 2 (SARS-CoV-2). Coronavirus disease-19 (COVID-19) is an infectious disease caused by SARS-CoV-2 [3] that is rapidly evolving affecting individuals of all ages, thus creating an ongoing global health crisis. SARS-CoV-2 enters host cells via angiotensin-converting enzyme 2 (ACE2) which is constitutively expressed by epithelial cells in various organs. Its expression in the terminal ileum and colon are amongst the highest in the body [4]. Although most cases of COVID-19 are mild, the disease can become severe and result in hospitalization, acute respiratory distress syndrome (ARDS) or multi-organ failure and death [3,5]. Data from recent waves of COVID-19 infections suggest that patients with underlying comorbidities are at greater risk to be infected by SARS-CoV-2 [6,7]. Immunosuppressed patients with IBD are included in this group. Nevertheless, in a recent large cohort of 5302 IBD patients only 39 (0.7%) developed COVID-19 [8]. In another population-based study, patients with IBD were more likely to be hospitalized due to COVID-19; however, the risk of severe COVID-19 was not higher than the general population [9]. A panel of the international organization for the study of inflammatory bowel diseases (IOIBD) has recently stated that having IBD did not increase the risk of developing COVID-19 [10,11]. However, data on the clinical course of COVID-19 among IBD patients especially in regard to immunosuppressive treatments remains relatively scarce. In the current work we prospectively collected data from COVID-19 patients with IBD and analyzed their outcomes with regard to their IBD treatment.

## 2. Materials and Methods

### 2.1. Study Setting

This nation-wide collaborative registry was initiated by the Israeli IBD Society at the start of the COVID-19 pandemic in March 2020, in order to monitor outcomes of COVID-19 among adult and pediatric patients with IBD. Seventeen IBD referral units in academic and non-academic hospitals and health centers affiliated with the Israeli IBD Society participated in the study. Physicians were prompted to report cases of PCR-confirmed COVID-19 occurring in patients with IBD, regardless of IBD treatment or infection severity. Patients were enrolled from March through October 2020.

### 2.2. Participants

Patients with established CD or UC were enrolled in 17 participating IBD centers. For all eligible patients, de-identified data from medical charts were collected, including age, gender, IBD type, IBD clinical activity, IBD treatments and comorbidities. UC and CD were defined according to the Montreal classification [12]. In UC patients, disease extension was classified to E1 (proctitis), E2 (left-sided) or E3 (extensive UC). The severity was classified as S0 (asymptomatic), S1 (mild), S2 (moderate) and S3 (severe). In CD patients, location was classified as L1 (ileal), L2 (colonic), L3 (ileocolonic) or L4 (isolated upper disease). Age at diagnosis was classified as A1 (below 16 years), A2 (between 17 and 40 years) and A3 (above 40 years). Behavior was classified as B1 (non-stricturing, non-penetrating), B2 (structuring) or B3 (penetrating).

### 2.3. Outcomes and Definitions

The primary objective of this study was to describe COVID-19 characteristics in patients with IBD. COVID-19 was defined as a positive PCR test for SARS-CoV-2 on a nasopharyngeal swab. For COVID-19 symptoms we used the definitions of the NIH for illness categories: asymptomatic infection, moderate, severe and critical illness [13]. In the present study the characteristics are described in terms of severe outcomes, such as hospitalization, need for respiratory support, intensive care and mortality. The secondary objective was to investigate possible associations between severe infection outcomes, IBD phenotypes and immunosuppressive/biologic treatments.

### 2.4. Research Ethics and Patient Consent

The study was approved on February 2020 by the institutional review board (no: 0020-19) of Shaare Zedek Medical Center (SZMC), as well as by the Ministry of Health Ethics Committee. All participants in the study had approval of their institution as well. The study protocol conforms to the ethical guidelines of the 1975 Declaration of Helsinki as reflected in a priori approval by the institutional human research committee. Participating centers reported de-identified data, in accordance with the HIPAA De-Identification standards.

### 2.5. Statistical Analysis

Categorical variables were compared using Chi-square test or Fisher’s exact test. Continuous variables were analyzed by Wilcoxon signed ranks test or paired samples t-test, according to variable distribution. All tests were two-tailed and significance was defined as *p*-value < 0.05. The data were analyzed using software package for statistics (IBM, SPSS version 25).

## 3. Results

### 3.1. IBD Patient Characteristics

Between March and October 2020, 144 patients with an established IBD diagnosis and confirmed COVID-19 were enrolled at 20 IBD referral units. Patient demographic and clinical characteristics are shown in Table 1. Most patients were under the age of 40 (113, 78.4%); 9 were under the age of 18 (6.3%) and only 4 were above the age of 70 (2.8%). In total, 104 had CD (72.2%) and 40 (27.8%) had UC. Mean age was 32.2 ± 12.6 years. Obesity was reported in 17 (11.8%) patients. Other comorbidities included diabetes (5.6%), asthma or chronic obstructive pulmonary disease (COPD) (4%) and congestive heart failure (CHF) (1.4%). The cohort included 14% tobacco smokers and 10% cannabis users (either oil or smokers). IBD characteristics were based on the Montreal classification [12]. For CD patients, 35.6% were A1, 53.8% A2 and 7.7% A3. The location was ileal (L1) in 43.3%, followed by ileocolonic (L3) 36.5%, and colonic (L2) 17.3%. A total of 8.7% of patients also presented with isolated upper disease (L4). Neither disease location nor behavior or activity correlated with the severity of COVID-19 infection. Additionally, 16.3% of CD patients and 30% of UC patients were infected by SARS-CoV-2 while in remission, the rest were reported with moderate disease.

IBD treatments are summarized in Table 2. Eighteen subjects did not receive any treatment. Ninety-four patients (65.3%) received biologics, either as a monotherapy (52.8%), or combined with immunomodulators (IM) (11%) or with concomitant corticosteroids (10%). One patient received biologics combined with IM and corticosteroids. Three patients received Tofacitinib and four patients were on a clinical trial medication (either new biologic or small molecule). Fifty-one (51%) patients received biologics at escalated doses. In a third of the patient population (34.7%) the treatment was postponed until recovery from COVID-19, with no resulting IBD exacerbation.

### 3.2. COVID-19 Characteristics in IBD Patients

Reported COVID-19 symptoms are summarized in Table 3. CD patients were more prone to severe COVID-19 than UC patients (*p* = 0.08). Most patients presented with mild disease and were treated at a home setting (*p* = 0.747). In addition, there was no correlation between IBD severity (as presented in Table 1) to COVID-19 severity, when divided into CD and UC (*p* = 0.49 and *p* = 0.7 for CD and UC, respectively). There was no significant difference between CD and UC groups in any symptom, except for shortness of breath that was more frequent in subjects with UC (*p* = 0.009). As for the GI symptoms, there was also no apparent aggravation of diarrhea (defined by before/after reported number of bowel movements). A small percentage suffered from vomiting or nausea. Overall, 24 patients (16.6%) were admitted for hospitalization, the rest were managed either by HMO (health maintenance organization) teams in home hospitalization set up (114, 76.2%), or by Home Front Command medical teams in hotels converted into makeshift healthcare facilities. Three hospitalized patients required endotracheal intubation and mechanical ventilation (2%), while 15 (10.4%) required non-invasive ventilation and oxygen support. There were no mortalities. All patients recovered from the infection uneventfully. As expected, age over 50 was significantly correlated to severe disease (*p* = 0.001).

### 3.3. Biological Treatments for IBD May Be Associated with Favorable Course of COVID-19

We tested whether any IBD treatment is correlated to COVID-19 severity. Interestingly, we found that overall patients that received biologics had a significantly milder course of disease (*p* = 0.001, Figure 1A). This finding remained significant following multivariate regression that took into account other treatments, as well as the type of disease, age over 50, gender, smoking status, obesity and comorbidities (*p* = 0.016, OR = 0.07, CI 95%: 0.009–0.621). In addition, more patients in the no-biologics group were hospitalized (*p* < 0.001, Figure 1B). When testing sub-groups of biologics, we found that among 52 patients treated with anti TNF, only 4 patients (7.7%) were hospitalized. Disease severity was significantly (*p* = 0.021), lower in this subgroup of patients treated with anti TNF, with only one severe and one moderate patient. Notably, 61% (11/18) of patients who did not receive any IBD treatment suffered from moderate or severe symptoms.

## 4. Discussion

Patients with IBD are primarily treated by immunosuppressive medications. Thus, during the outbreak of COVID-19 there was increasing concern regarding their risk of being infected with SARS-CoV-2. Accumulating data from the last 2 years found no evidence for an increased susceptibility to COVID-19 among patients with IBD [9,10,14,15,16,17]. In our prospective observational study, we found that IBD patients treated with biological medications even had a protecting effect from severe SARS-CoV-2 infection. Similarly, one of the first observations of SARS-CoV-2 from Wuhan, China did not report any COVID-19 case among 318 IBD patients [18]. Anti-TNF antibodies are frequently used for IBD therapy [19,20,21]. Our cohort included 52 patients who received anti-TNF treatments and experienced only a mild course of COVID-19. These results are in line with the decreased risk for hospitalization reported in anti TNF-treated patients with other rheumatic diseases who were infected with COVID-19 [22]. The expression of ACE2 is increased in the inflamed gut of patients with IBD [23]. On the other hand, the level of soluble ACE2, a different form of this protein that lacks the membrane anchor and circulates in small amounts in the blood, is also upregulated in the blood of patients with IBD [24]. Therefore, soluble ACE2 might be a competitive interceptor for SARS-CoV-2 by preventing the binding of the viral particle to the surface-bound, full length ACE2 [25]. These observations suggest a limited infection rate for IBD patients. Mehta et al. showed that TNF may also exert pathogenic effects by augmenting the expression of ACE2 or by augmenting lymphopenia through induction of direct leucocyte death via T cell TNF or TNF receptor signaling [26]. Furthermore, Burgueno et al. have recently showed that the expression of ACE2 was not increased in patients with IBD and further therapy with biologics may decrease the expression of these receptors, resulting in overall support for a milder course in IBD patients [27]. Moreover, a new trial recently launched will investigate whether adalimumab, an anti-TNF drug, is effective for treating COVID-19 patients in the community [28]. Our data supports these reports, as we show favorable outcomes in our cohort, which included 31 patients receiving adalimumab. Taken together, the favorable outcome observed in IBD patients treated with anti TNFs support a potentially protective effect of TNF inhibition in COVID-19, but further investigations are required.

Vedolizumab, a selective α4β7 inhibitor, and Ustekinumab, an IL12/23 inhibitor, are usually given to IBD patients who have failed or who are intolerant of anti-TNF therapy. Both drugs are approved for IBD and no increase in viral infections has been noted in patients treated with these agents. Vedolizumab was given to 23 patients and Ustekinumab was given to 12 patients in our study, none of them experienced severe course of COVID-19. Thiopurine (azathioprine, 6-mercaptopurine) and JAK1/3 inhibitor (tofacitinib) treatments can potentially reduce the number of activated T cells and affect T-cell activation and effector function. Although no data are currently available in this context, it might affect the course of COVID-19, as lymphopenia was associated with worse prognosis in this disease. Moreover, tofacitinib was reported to increase the risk for certain viral infections (e.g., herpes zoster) [29,30]. Since our cohort included only seven patients on tofacitinib, it is difficult to draw any specific conclusions. However, we can report that only one patient experienced a severe course of infection. In contrast to biologic drugs, corticosteroids which are also very common in treating IBD were found to be associated with adverse COVID-19 outcomes. The SECURE-IBD is a large international registry that was created to monitor outcomes of IBD patients with confirmed COVID-19. The risk factors for severe COVID-19 among the 525 IBD patients published by Brenner et al. included older age, increasing number of comorbidities and systemic corticosteroids. Similarly to our results, they reported that anti TNF agents were not associated with severe COVID-19 and patients on TNF antagonists were younger and more likely to have CD [31]. Notably, our study only included eight patients with severe course of COVID-19, with no deaths. Two of them were treated with biologics and two with steroids. Therefore, our findings do not support the reported increased risk in corticosteroid use.

Smoking is most likely associated with negative progression and severe outcomes of COVID-19 [32,33]. Our cohort included a small percentage of smokers, resulting in no significant correlation to COVID-19 severity, although a trend was observed in the multivariate analysis (*p* = 0.25, OR = 4, CI 95%: 0.37–0.42).

A study limitation is the relatively small number of patients in the cohort. Therefore, there was no specific analysis regarding the dosage or the duration of each treatment prior to COVID-19 onset. However, IBD patients were enrolled from all over the country, which is a major strength of the study, since it makes the results more generalizable. The study was performed in the first few months of the COVID-19 pandemic, including low number of patients treated with corticosteroids that also emphasizes the native perspective of the disease without SARS-CoV-2 variants and still without the effect of vaccination.

In summary, our study supports recent reports showing that there is no evidence for aggravated outcomes in patients with IBD in the context of COVID-19. Therefore, patients should probably continue their scheduled IBD-specific medications. Moreover, our findings may suggest a favorable outcome for patients receiving biologic treatment. However, in severe cases in which experimental medications are given (e.g., remdesivir), drug–drug interactions should be considered.

## Figures and Tables

**Figure 1 vaccines-10-00376-f001:**
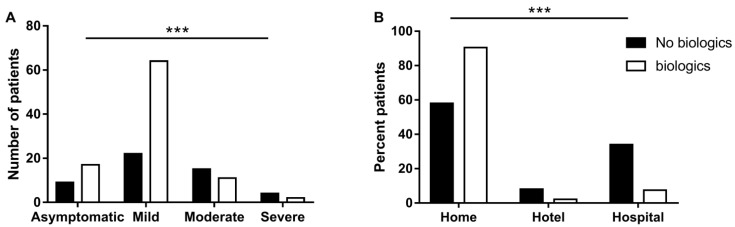
Association between IBD treatment and COVID-19 severity. (**A**) Number of IBD patients with various degrees of COVID-19 severity comparing biological therapy and non-biological therapy. (**B**) Outcome of COVID-19 in patients with IBD, comparing biological therapy and non-biological therapy. The *p*-value is based on a Pearson Chi-square test comparing patient distribution between groups (biological therapy and non-biological therapy), with *p* < 0.001 (***) in all degrees of COVID-19 severity and outcomes.

**Table 1 vaccines-10-00376-t001:** Patient characteristics.

	CD (*n* = 104)		UC (*n* = 40)		*p*
Gender (% Male)	46 (44.2%)		18 (45%)		0.934
**Montreal classification**					
	B1	69 (66.3%)	E1	4 (10%)	
	B2	23 (22.1%)	E2	12 (30%)	
	B3	12 (11.5%)	E3	20 (50%)	
	perianal disease	21 (20.2%)	colectomy	4 (10%)	
**IBD disease severity**					
Remission/Asymptomatic	17 (16.3%)		12 (30%)		*p* < 0.001
S1—Mild	51 (49%)		6 (15%)		
S2—Moderate	25 (24%)		10 (25%)		
S3—Severe	11 (10.6%)		8 (20%)		

**Table 2 vaccines-10-00376-t002:** IBD treatments in the patient cohort.

Treatment	Total (*n*)	Mild/Asymptomatic	Moderate/Severe	*p*
Anti-TNF	52	50	2	*p* = 0.002
Vedolizumab	23	22	1	*p* = 0.084
Ustekinumab	12	10	2	1
Other *	7			
Total biologics	94	87	7	**<0.001**
Biologics only	76	71	5	0.001
Biologics + steroids	9	7	2	0.644
Biologics + IM	10	10	0	0.143
IM monotherapy	12	11	1	0.418
Steroids only	8	7	1	0.745
IM monotherapy + steroids	2	2	0	0.524
5-ASA	29	25	4	0.642
none	18	7	11	**<0.001**

* Other included tofacitinib and trial biologics treatments. IM; immunomodulators, including methotrexate.

**Table 3 vaccines-10-00376-t003:** COVID-19 characteristics according to CD and UC populations.

IBD	CD (*n* = 104)	UC (*n* = 40)	*p* Value
Asymptomatic	25 (24%)	5 (12.5%)	0.08
Mild	64 (61.5%)	26 (65%)	
Moderate	8 (7.7%)	8 (20%)	
Severe	7 (6.7%)	1 (2.5%)	
**Setting**			
Home	84 (80.8%)	30 (75%)	0.747
Hotel	4 (3.8%)	2 (5%)	
Hospital	16 (15.4%)	8 (20%)	
**Symptoms for COVID-19**			
Fever	49 (47.1%)	16 (40%)	0.442
Cough	43 (41.3%)	15 (37.5%)	0.673
Shortness of breath	9 (8.7%)	10 (25%)	**0.009**
Fatigue	9 (8.7%)	3 (7.5%)	0.822
Headache	9 (8.7%)	5 (12.5%)	0.485
Dysgeusia	13 (12.5%)	5 (12.5%)	1
Throat ache	4 (3.8%)	3 (7.5%)	0.361
GI Pain	19 (18.3%)	8 (20%)	0.3
Vomiting	5 (4.8%)	2 (5%)	0.693
Nausea	2 (1.9%)	1 (2.5%)	0.65
Diarrhea *	11 (10.5%)	6 (15%)	0.46

* Diarrhea was defined as more than 5 daily bowel movements.

## Data Availability

The data presented in this study are available on request from the corresponding author. The data are not publicly available fur to ethical restrictions.

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
