# Peer review of "COVID-19 in Patients with Inflammatory Bowel Disease: The Israeli Experience"

_vaccines, 2022, doi:10.3390/vaccines10030376_

Round 1

Reviewer 1 Report

COVID-19 IN PATIENTS WITH INFLAMMATORY BOWEL DISEASE: THE ISRAELI EXPERIENCE

Crohn’s disease and ulcerative colitis are chronic, imune mediated inflammatory bowel diseases (IBD) affecting millions of people worldwide. IBD therapies, designed for continuous immune suppression, often render patients more susceptible to infections.

The present work is quite pertinent and interesting because the effect of the immune suppression on the risk of coronavirus disease-19 (COVID-19) is not fully determined yet.

The authors do a thorough discussion of the results well supported by the existing literature in the area.

Notes:

P1 – I suggest that the objective defined in the abstract must be broader, to suit the title of the article and the objectives defined in section 2.3. “Outcomes and definitions”

P2 - Keywords must not include abbreviations.

P3 – In Table 1, it will be interesting to make 2 lines to separate the masculine gender from the feminine gender.

P4, line 145 - Leave 1 line to separate table 1 from the text.

P4, line 147 - Abbreviation “IM” is not described in full the first time it is mentioned. This abbreviation is only described in the table further on, in the legend, line 154.

P4, line 150-151 - “In one third of the patient population (50, 34.7%)”. I suggest putting only the percentage value in parentheses and removing the 50.

P5 – Figure 1A and 1B should be a legend, as well as an explanation of the meaning of the ***.

Author Response

Comments and Suggestions for Authors

COVID-19 IN PATIENTS WITH INFLAMMATORY BOWEL DISEASE: THE ISRAELI EXPERIENCE

Crohn’s disease and ulcerative colitis are chronic, immune mediated inflammatory bowel diseases (IBD) affecting millions of people worldwide. IBD therapies, designed for continuous immune suppression, often render patients more susceptible to infections. 

The present work is quite pertinent and interesting because the effect of the immune suppression on the risk of coronavirus disease-19 (COVID-19) is not fully determined yet. The authors do a thorough discussion of the results well supported by the existing literature in the area.

Notes:

P1 – I suggest that the objective defined in the abstract must be broader, to suit the title of the article and the objectives defined in section 2.3. “Outcomes and definitions”

- Corrected

P2 - Keywords must not include abbreviations.

- Corrected

P3 – In Table 1, it will be interesting to make 2 lines to separate the masculine gender from the feminine gender.

- Corrected

P4, line 145 - Leave 1 line to separate table 1 from the text.

- Corrected

 P4, line 147 - Abbreviation “IM” is not described in full the first time it is mentioned. This abbreviation is only described in the table further on, in the legend, line 154.

- Corrected

P4, line 150-151 - “In one third of the patient population (50, 34.7%)”. I suggest putting only the percentage value in parentheses and removing the 50.

- Corrected

 P5 – Figure 1A and 1B should be a legend, as well as an explanation of the meaning of the ***.

- Corrected

Reviewer 2 Report

The article presented by Lev Lichtenstein and collaborates, entitled “COVID-19 in patients with inflammatory bowel disease: The Israeli experience”, is a multi-center study that evaluate the effect of immunomodulatory therapies on the clinical course of COVID-19 in IBD patients. The article includes 144 patients from Israel. The article is well written, easy to understand and the vocabulary is correct. In general, the work does not go too deeply into the correlation between the administration of immunomodulators in patients with inflammatory bowel disease and the symptomatology produced by COVID19. In addition, the work has serious faults:

Major revision: 

First, Table 3 is missing and without the table it is not possible to understand the work and more importantly, to draw conclusions

Second, Figure 1 is totally incorrect, it has no caption explaining the axes, no errors, no number of patients per group, etc.

 Minor revision:

Table 1 divided in two

Author Response

Comments and Suggestions for Authors

The article presented by Lev Lichtenstein and collaborates, entitled “COVID-19 in patients with inflammatory bowel disease: The Israeli experience”, is a multi-center study that evaluate the effect of immunomodulatory therapies on the clinical course of COVID-19 in IBD patients. The article includes 144 patients from Israel. The article is well written, easy to understand and the vocabulary is correct. In general, the work does not go too deeply into the correlation between the administration of immunomodulators in patients with inflammatory bowel disease and the symptomatology produced by COVID19. In addition, the work has serious faults:

Major revision: 

First, Table 3 is missing and without the table it is not possible to understand the work and more importantly, to draw conclusions

Table 3 was not missing and was attached with the manuscript

Second, Figure 1 is totally incorrect, it has no caption explaining the axes, no errors, no number of patients per group, etc.

- Corrected

 Minor revision:

Table 1 divided in two

- Corrected

Round 2

Reviewer 2 Report

The authors have introduced some of the changes suggested in the first revision, but some errors have not been corrected.

  1. Table 1 continues to be divided in different pages
  2. In Figure 1 the statistics are not clear. It is not possible to know which groups are significant with respect to which particular group. The way of expressing the statistics should be improved, both in the graph and in the figure caption, and not just put a line with ***.

Author Response

Now table 1 is not divided between 2 pages and appears on page 4, line 146.

Regarding the legend of Figure 1: We added the following sentence in the legend, hoping that now it is clearer:

The p-value is based on a Pearson chi- square test comparing patient's distribution between groups (biological therapy and non-biological therapy), with p < 0.001 (***) in all degrees of COVID-19 severity and outcomes.